# Self-Attention-Augmented Generative Adversarial Networks for Data-Driven Modeling of Nanoscale Coating Manufacturing

**DOI:** 10.3390/mi13060847

**Published:** 2022-05-29

**Authors:** Shanling Ji, Jianxiong Zhu, Yuan Yang, Hui Zhang, Zhihao Zhang, Zhijie Xia, Zhisheng Zhang

**Affiliations:** 1The School of Mechanical Engineering, Southeast University, Nanjing 211189, China; slji@seu.edu.cn (S.J.); yangyuancsi@163.com (Y.Y.); zhanghuihui@seu.edu.cn (H.Z.); 220204830@seu.edu.cn (Z.Z.); 2State Key Laboratory of Virtual Reality Technology and Systems, Beihang University, Beijing 100191, China; 3State Key Laboratory of Transducer Technology, Chinese Academy Sciences, Shanghai 200050, China

**Keywords:** data-driven modeling, generative adversarial network, nanoscale coating manufacturing, self-attention

## Abstract

Nanoscale coating manufacturing (NCM) process modeling is an important way to monitor and modulate coating quality. The multivariable prediction of coated film and the data augmentation of the NCM process are two common issues in smart factories. However, there has not been an artificial intelligence model to solve these two problems simultaneously. Focusing on the two problems, a novel auxiliary regression using a self-attention-augmented generative adversarial network (AR-SAGAN) is proposed in this paper. This model deals with the problem of NCM process modeling with three steps. First, the AR-SAGAN structure was established and composed of a generator, feature extractor, discriminator, and regressor. Second, the nanoscale coating quality was estimated by putting online control parameters into the feature extractor and regressor. Third, the control parameters in the recipes were generated using preset parameters and target quality. Finally, the proposed method was verified by the experiments of a solar cell antireflection coating dataset, the results of which showed that our method performs excellently for both multivariable quality prediction and data augmentation. The mean squared error of the predicted thickness was about 1.6~2.1 nm, which is lower than other traditional methods.

## 1. Introduction

Nanoscale coating technology is widely used in the advanced industrial manufacturing, such as solar cell antireflection films and new multifunctional materials for the automobile and aircraft industries [1]. The process modeling of nanoscale coating manufacturing (NCM) can be utilized to predict coating quality and analyze the effects of coating control parameters (recipes). However, NCM processes, including coating processes using chemical vapor deposition (CVD), dip-coating, sputtering, and other methods [2], are usually complex, nonlinear processes that are difficult to model. In addition, advanced data-driven models can provide recipe guidance by using data augmentation in industrial manufacturing. Therefore, the need for improving coating quality has necessitated more and more intelligent applications of NCM process modeling for quality prediction.

In the literature, coating process modeling methods can be classified into statistical-model-based methods and artificial-intelligence-based methods [3,4]. Response surface methodology [5,6], analysis of variance [7], the finite element method [8], the Taguchi design method [9,10], and other statistical analyses [11,12] are frequently used statistical methods. However, statistical-model-based methods have the limitation of subjectively selecting coating control conditions by executing the designs of experiments. Moreover, conventional statistical methods are not suitable for complex multivariate nonlinear NCM process control in industrial manufacturing. Artificial intelligence methods such as machine learning (ML) and deep learning (DL) are more suitable for handling data-driven process modeling and solving nonlinear problems. For example, typical control factors have been fed to machine learning models such as a support vector machine (SVM) [13], a neural network (NN) [14], or a Gaussian process regression (GPR) [15] to predict coating quality. Paturi et al. [16] employed a genetic algorithm (GA) and response surface methodology to establish the optimum conditions for electrostatic spray deposition parameters, and they estimated coating thickness using proposed artificial neural network (ANN) and SVM models. However, this hybrid method had significant cost for model training and could not ensure production fluctuation. Recently, DL methods also supplied an end-to-end learning approach for NCM process modeling and quality prediction [17].

Generally, the defects in existing methods are summarized by following aspects:(1)The relations among different manufacturing steps are ignored when extracting features from the control recipes;(2)Data augmentation is an essential technique in DL-based process modeling in industrial manufacturing. Prior works have few studies about recipe augmentation, especially in NCM;(3)The multivariable quality prediction and data augmentation of NCM are rarely considered simultaneously, as these factors can increase the training cost.

From a data-mapping perspective, NCM process modeling helps to establish the relationship between coating quality and corresponding recipes. Nevertheless, most research works have only studied modeling for coating quality prediction in which the input variables have been recipes, and the output variables have been coating quality factors. Modeling of coating recipe generation for desired quality or specific control conditions has been ignored. Theoretically, if recipe generation for particular quality factors is required, a model can be obtained by inverting the coating quality prediction model. In addition, the latent coupling information between the post-process recipes and the pre-process recipes is beneficial for the control of the multilayer coating process. Generative adversarial networks (GANs) provide the possibility of complete NCM processing modeling.

Self-attention generative adversarial networks (SAGANs) [18] inspire establishing NCM processing models for quality prediction and data augmentation. In an improved SAGAN, an additional regressor in parallel with a discriminator is exploited to predict multivariable quality factors while a generator is applied for data augmentation assisted by a self-attention mechanism. Therefore, an auxiliary regression using a self-attention-augmented generative adversarial network (AR-SAGAN)-based NCM data-driven process model is proposed. 

The major novelties and contributions of this paper can be summarized in three aspects.

(1)A data-driven NCM process model is proposed in an end-to-end way that can predict coating quality by learning features adaptively from complex industrial process data and can make data augmentation by generating recipes of coating processing.(2)The data augmentation of the multilayer coating processing is challenging work. The proposed model not only learns the connection information between the NCM output quality and the control parameters, but it also extracts latent knowledge between the former coating steps and the subsequent coating steps from history production data with the assistance of a self-attention technique.(3)The quality of the NCM output has multiple variables, which may include thickness, refractive index, or other reference values. In addition, there is a coupling relationship between these output values. The proposed framework can predict multivariable quality by sharing feature information of control parameters and regression weights.

The rest of this paper is organized as follows. The preliminaries of NCM process modeling, as well as the self-attention mechanism and basic GAN, are described in Section 2. Section 3 illustrates the proposed AR-SAGAN function and its training algorithm in detail. In Section 4, the proposed method is applied to analyze a dataset of an NCM instance. The results and comparisons with different regression variables and other methods verify the effectiveness of the proposed AR-SAGAN framework. Finally, Section 5 concludes this paper.

## 2. Background Knowledge

In this section, background knowledge of NCM process modeling using ANNs, as well as self-attention mechanisms and generative adversarial networks, is demonstrated.

### 2.1. NCM Process Modeling Using ANNs

ANNs have been proved for the application of coating process quality prediction, especially coating thickness estimation [19]. The structure of a typical ANN is shown in Figure 1a. The hidden layers that connect the input and output layers include computable nodes. The input and output vectors of each layer in the network can be obtained by forward layer-by-layer calculation. Through error back-propagation, the loss is calculated, and the network is updated.

### 2.2. Self-Attention Mechanism

A self-attention mechanism is used to connect and capture correlations among different vectors [20]. They have been used for fault detection and diagnosis in semiconductor manufacturing [21]. However, self-attention-augmented data augmentation and feature extraction in NCM have not been studied. The self-attention module utilized in this study is displayed as Figure 1b. The query, keys values, and output can be obtained from the same inputs through different linear layers. Using a self-attention mechanism means a query and a set of key-value pairs are projected to an output. The queries, keys, and values are concatenated into matrices Q, K, and V to parallelize the calculations. The output of self-attention can be expressed as:(1)$$\mathrm{Attention}\left(Q,K,V\right)=\mathrm{SoftMax}\left(QK^{T}\right)V$$

### 2.3. Basic Generative Adversarial Networks

The basic generative adversarial network (GAN) proposed by Goodfellow et al. [22] is composed of a discriminator *D* and a generator *G*, which are both fully connected. The generator can take noise data and create fake data. The discriminator can distinguish between the fake data and real data. An auxiliary classifier GAN (ACGAN) [23] adds an extra classifier structure at the output end of the discriminator, as shown in Figure 1c. Thus, when training the discriminator and generator, the classifier is trained at the same time. In addition to generator and discriminator losses, classification losses are also considered when calculating training losses. Therefore, the ACGAN can generate images with a conditional image label. However, most GAN-related studies are generally related to image synthesis and classification. In a previous work, the continuous labels were quantized to limited classes [24], which is not suitable for continuous variable prediction with subtle tolerance.

Herein, NCM process modeling using GAN is responsible for satisfying three key points: (1)The generated data for target coating quality;(2)The discriminator to distinguish between real control parameters and generated parameters;(3)The regression for quality estimation using the input control parameters.

Inspired by ACGAN, the aforementioned improved GAN was defined as AR-SAGAN (auxiliary regression using SAGAN).

## 3. Proposed Approach

In order to model the NCM process and solve quality prediction and augmented recipe generation synchronously, an AR-SAGAN architecture was proposed. Figure 2 illustrates the overview of NCM quality prediction and data augmentation using AR-SAGAN, which mainly consisted of four steps.

(1)Preprocessing. The collected data included control data and associated quality data. In addition to deposition time, the raw control data sampled from multiple sensors were continuous and fluctuated around the original control value. Thus, the median values in each coating step were extracted as the feature. After that, outlier elimination and normalization were carried out.(2)Model training. Our proposed AR-SAGAN was trained using an offline dataset. The AR-SAGAN was periodically trained and updated to adapt the real-time operating conditions.(3)Quality prediction. The online control parameters were collected, preprocessed, and then input to a regressor, which was trained using AR-SAGAN to predict quality.(4)Data augmentation. In this step, the online control parameters and the target quality data were preprocessed and input to a generator trained by AR-SAGAN to generate more control recipes.

### 3.1. AR-SAGAN Model

The specific AR-SAGAN model architecture is depicted in Figure 3. The model architecture of AR-SAGAN was mainly divided into four parts: a generator (as shown in Figure 3a), a feature extractor, a discriminator, and a regressor (as shown in Figure 3b). A self-attention module independently extracted latent correlations between different control parameters in the generator and feature extractor. Concretely, the roles and connections of the different parts were demonstrated as follows.

(1)The generator took random noise, desired quality data, and control parameters of the first λ coating steps as the input. Subsequently, the implied feature of the control parameter matrix was concatenated with quality data and noise via the self-attention module. The output of generator was the last m−λ steps of the recipe. Finally, to output the complete recipe, a concatenation operation was employed between the control parameters of the first λ steps and the generated last m−λ steps.(2)The feature extractor extracted latent information from the complete recipe. The control parameters were reshaped into the size m×n and then passed through the self-attention module. The module output was connected with a flattened layer, which was related with the discriminator and regressor. The discriminator distinguished between the real recipe or fake recipes (generated control parameters). The regressor predicted the coating quality based on the complete coating recipe.

### 3.2. Loss Function

To train the AR-SAGAN model, the losses were defined, including discriminator loss $L_{D}$, generator loss $L_{G}$, and regressor loss $L_{REG}$. According to the game model of the GAN, the optimization condition was a minimized generator loss and a maximized discriminator loss. In addition, the regressor loss was minimized. Therefore, the objective function of AR-SAGAN was:(2)$$\underset{REG}{\min}\;\underset{G}{\min}\;\underset{D}{\max}L\;\left(REG,G,D\right)$$

Due to multivariable outputs, the regressor loss was hybrid. For *N* data pairs, the mean absolute error (MAE) $\left|{\hat{y}}_{i}-y_{i}\right|/N$ was implemented between the real data y and the predicted data $\hat{y}$, which was calculated as:(3)$$L_{REG}=\overset{l}{\underset{i=1}{\sum }}w_{i}\mathrm{MAE}\left({\hat{y}}_{i},y_{i}\right)$$
where $w_{i}$ is the loss weight.

The Wasserstein-distance-based GAN (WGAN) is proven to be more suitable for stability training compared with using KL divergence and JS divergence [25]. To ensure the Lipschiz continuity of the critic, WGAN is improved with a gradient penalty (WGAN-GP) [26]. Therefore, the loss function of WGAN-GP was adopted to calculate discriminator loss:(4)$$L_{D}=\underset{\tilde{x}\sim P_{fake}}{E}\left[D\left(\tilde{x}\right)\right]-\underset{x\sim P_{real}}{E}\left[D\left(x\right)\right]+w\underset{\hat{x}\sim P_{\hat{x}}}{E}\left[{\left(\nabla_{\hat{x}}D{\left(\hat{x}\right)}_{2}-1\right)}^{2}\right]$$
where $\hat{x}=\epsilon \tilde{x}+\left(1-\epsilon \right)x$, and w is the weight of the gradient penalty loss. Then, maximizing the discriminator loss results in minimizing $L_{D}$.

The generator loss evaluated the generated fake data based on Wasserstein distance:(5)$$L_{G}=-\underset{\tilde{x}\sim P_{fake}}{E}\left[D\left(\tilde{x}\right)\right]$$

### 3.3. Training Algorithms

$\theta_{G}$, $\theta_{F}$, $\theta_{D}$, and $\theta_{RGE}$ represent the learnable parameters of the generator, feature extractor, discriminator, and regressor, respectively. To make the training convergence, the discriminator was trained first for the k loops, and then the generator and regressor were trained. 

Because the discriminator and regressor shared the weights and parameters of the feature extractor, there were three training conditions to update the feature extractor. Training condition 1 (TC1) was for training the feature extractor based on the discriminator loss and to then freeze the weights of the feature extractor to train the regressor. In the case of TC1, the learnable parameters were updated as follows:(6)$$\left({\hat{\theta }}_{F},{\hat{\theta }}_{D}\right)=\underset{\theta_{F},\theta_{D}}{\mathrm{argmin}}\;L_{D}\left(\theta_{G},\theta_{F,}\theta_{D}\right)\;$$
(7)$${\hat{\theta }}_{G}=\underset{\theta_{G}}{\mathrm{argmin}}\;L_{G}\left(\theta_{G},{\hat{\theta }}_{F,}{\hat{\theta }}_{D}\right)$$
(8)$${\hat{\theta }}_{REG}=\underset{\theta_{REG}}{\mathrm{argmin}}\;L_{REG}\left({\hat{\theta }}_{F,}\theta_{REG}\right)$$

The learning algorithm of the AR-SAGAN model based on TC1 is summarized in Algorithm 1.
**Algorithm 1**: Training AR-SAGAN based on TC1.**Input**: Preal={Xim,Yi}i=1Nr, Pfake={Xiλ,Yif,Zi}i=1Nf Initialize network parameters {θG, θF, θD, θRGE}**while** not converged **do**
**for**
*k* steps **do**
$\nabla_{\theta_{F},\theta_{D}}L_{D}\left(X^{m},X^{\lambda },Y^{f},Z\right)$
**end**
∇θGLG(Xλ,Yf,Z)∇θREGLREG(Xm,Y)**end while**

Training condition 2 (TC2) always updated the weights of the feature extractor based on the discriminator loss and regressor loss. In the case of TC2, the parameters were updated as follows:(9)$$\left({\hat{\theta }}_{F},{\hat{\theta }}_{D}\right)=\underset{\theta_{F},\theta_{D}}{\mathrm{argmin}}\;L_{D}\left(\theta_{G},\theta_{F,}\theta_{D}\right)$$
(10)$${\hat{\theta }}_{G}=\underset{\theta_{G}}{\mathrm{argmin}}\;L_{G}\left(\theta_{G},{\hat{\theta }}_{F},{\hat{\theta }}_{D}\right)$$
(11)$$\left({\hat{\hat{\theta }}}_{F},{\hat{\theta }}_{REG}\right)=\underset{{\hat{\theta }}_{F},\theta_{REG}}{\mathrm{argmin}}\;L_{REG}\left({\hat{\theta }}_{F},\theta_{REG}\right)$$

The learning algorithm of the AR-SAGAN model based on TC2 is summarized in Algorithm 2.
**Algorithm 2**: Training AR-SAGAN based on TC2.**Input**$:\;P_{real}={\left\{X_{i}^{m},Y_{i}\right\}}_{i=1}^{N_{r}}$, $P_{fake}={\left\{X_{i}^{\lambda },Y_{i}^{f},Z_{i}\right\}}_{i=1}^{N_{f}}$ Initialize network parameters $\{\theta_{G}$, $\theta_{F}$, $\theta_{D}$, $\theta_{RGE}\}$
**while** not converged **do**
**for**
*k* steps **do**∇θF,θDLD(Xm,Xλ,Yf,Z)**end**∇θGLG(Xλ,Yf,Z)∇θF,θREGLREG(Xm,Y)**end while**

Training condition 3 (TC3) only trained the feature extractor using the regressor loss and froze the weights when training the discriminator. For the last case, the parameters were updated as follows:(12)$${\hat{\theta }}_{D}=\underset{\theta_{D}}{\mathrm{argmin}}\;L_{D}\left(\theta_{G},\theta_{F,}\theta_{D}\right)\;$$
(13)$${\hat{\theta }}_{G}=\underset{\theta_{G}}{\mathrm{argmin}}\;L_{G}\left(\theta_{G},\theta_{F,}{\hat{\theta }}_{D}\right)$$
(14)$$\left({\hat{\theta }}_{F},{\hat{\theta }}_{REG}\right)=\underset{\theta_{F},\theta_{REG}}{\mathrm{argmin}}\;L_{REG}\left(\theta_{F},\theta_{REG}\right)$$

The learning algorithm of the AR-SAGAN model based on TC3 is summarized in Algorithm 3.
**Algorithm 3:** Training AR-SAGAN based on TC3.**Input**$:\;P_{real}={\left\{X_{i}^{m},Y_{i}\right\}}_{i=1}^{N_{r}}$, $P_{fake}={\left\{X_{i}^{\lambda },Y_{i}^{f},Z_{i}\right\}}_{i=1}^{N_{f}}$ Initialize network parameters $\{\theta_{G}$, $\theta_{F}$, $\theta_{D}$, $\theta_{RGE}\}$
**while** not converged **do**
**for**
*k* steps **do**
$\nabla_{\theta_{D}}L_{D}\left(X^{m},X^{\lambda },Y^{f},Z\right)$
**end**
∇θGLG(Xλ,Yf,Z)∇θF,θREGLREG(Xm,Y)**end while**

## 4. Case Study

### 4.1. Experimental Setup and Dataset Description

Plasma-enhanced chemical vapor deposition (PECVD) is a coating technique with the auxiliary of radio frequency that promotes the formation of a gaseous reaction ionization environment, boosting the deposition rate of the film [27]. Silicon nitride (SiNx) thin films deposited using the PECVD process have excellent photoelectric and mechanical properties and are widely used in the coating of integrated circuits, micromechatronics, solar cells, and display devices. The SiNx thin-film deposition process using PECVD is illustrated in Figure 4a,b. Mixed gas including ammonia and silane is filled into the reaction chamber. With suitable reaction conditions, the ammonia reacts with silane in proportion to form silicon nitride precipitate [28]. After a period of deposition, the NCM thin-film thickness increases, and a corresponding refractive index is obtained. The gas amounts of ammonia and silane can be changed in different procedures to produce multilayer films with different properties. Although factories can use big data technology to record and analyze historical PECVD process data, there is no simple control model for quality prediction and automatically generated recipes of a desired quality.

The experimental data were sampled from a practical process consisting of 3 coating steps and 20 control parameters. The control parameters were sampled using multisensors, and the sampling frequency was 0.5 Hz. The average thickness (TN) and refractive index (RI) of the solar cells were measured using an ellipsometer after the coating process. The recorded ranges of TN and RI were 70~80 nm and 2.1~2.5, respectively. The control parameters included temperatures of different areas, cavity pressure, RF power, gas flow velocity, relative flow ratio among gases, and deposition time of each coating step. The variation trends of these control parameters are shown in Figure 4c–h. Before training and testing AR-SAGAN, preprocessing of the experimental data was implemented as described in Section 3. The control parameters and quality data were normalized into the range of [0, 1].

### 4.2. Performance of AR-SAGAN

The AR-SAGAN model was implemented, and the algorithms were compared under different training conditions. There were 500 training data and 183 test data. For the training data, the batch size for real data and fake data was 128. The first 2 coating steps were used as the input to output the 20 control parameters of the last coating step. The number of random noises was 1. An Adam learning optimizer was used. 

The mean absolute percentage error (MAPE) $\overset{N}{\underset{i=1}{\sum }}\left|y_{i}-{\hat{y}}_{i}\right|/y_{i}N$ was utilized to measure the distance between the generated control parameters and the real parameters. The epoch number was 100. The test metrics of the generated control parameters are shown in Table 1. It seemed that TC3-based training results had a lower error, followed by TC2. The feature extractor updated using real data made the generated data more stable.

The regressor was always trained with the real data. However, the parameters of the feature extractor were influenced by the fake data in the cases of TC1 and TC2. The normalized outputs of regressor were inversely transformed into the original ranges, and the metrics were calculated. The loss weights of the thickness and refractive index were 1 and 2, respectively. The epoch number was 200. In addition to the MAPE, mean squared error (MSE) $\overset{N}{\underset{i=1}{\sum }}{\left(y_{i}-{\hat{y}}_{i}\right)}^{2}/N$ was also utilized as the metrics. The predicted results of coating quality based on different training conditions are also compared in Table 2. The prediction errors based on TC2 and TC3 were lower than that based on TC1. Combined with the results of the generated control parameters in Table 1, the feature extractor updated using regression loss with real data improved the performance in data augmentation and quality prediction.

### 4.3. Practical Application in NCM

As shown in Figure 5, the data-driven process modeling of NCM was instructive in practical application. The processing data acquired from the physical manufacturing process and utilized for digital modeling. After that, a data mining technique is applied to obtain the production information. Meanwhile, the digital-driven model is trained. Furthermore, the data augmentation and quality prediction can be visualized in a virtual simulation, which can provide suggestions for manufacturing management in smart factories. For data augmentation, more control recipes can be generated, and then the operation formula can be adjusted according to practical production requirements. Moreover, the real-time data measured from the sensors and metrology can be simulated in a virtual space. For instance, the deposition schedule of an NCM process can be monitored instantly. Above all, there must be some product quality that is not measured in time but, instead, uses control parameters that can be collected easily. In this case, the quality of unlabeled products can be predicted using a data-driven model. Therefore, the AR-SAGAN model can be utilized in practical application to improve manufacturing management.

### 4.4. Comparison and Discussion

The compared regression results of AR-SAGAN and other methods are demonstrated in Table 3. The input and output of SVM are control parameters and coating quality. For CGAN [29], the generator takes in control parameters and noise and output predicted labels; then, the discriminator takes in control parameters and quality labels (predicted and real) and outputs the possibility of fake or real. The errors for training and tests using SVM, CGAN, and AR-SAGAN were compared. It can be seen that the thickness and refractive index were predicted, and the results of AR-SAGAN were better than the other methods.

From the aspect of architecture, the AR-SAGAN model was mainly composed of ANN and a self-attention module. Compared with other GAN-based models, AR-SAGAN not only controlled the labels of generated data, but it also estimated the continuous regression values at the regressor. Moreover, AR-SAGAN studied temporal characteristics by learning the latent relationships between the preset characteristics and generating follow-up information. The regressor included multiple-output branches and estimated labels using regression. The sample amount of random noise and preset information that was taken to the generator resolved the mode collapse for the target labels. Overall, the AR-SAGAN overcome quality prediction and data augmentation issues better than other conventional methods and can be applied in practical engineering.

## 5. Conclusions

To predict coating quality and augment recipe data for the NCM process in factories, this paper proposed a novel processing modeling method based on a self-attention mechanism and a GAN. First, the AR-SAGAN was proposed with data-driven auxiliary regression and self-attention-augmented adversarial generative structures. Furthermore, a case study on PECVD processing was provided to validate the effectiveness of the proposed AR-SAGAN. The results showed that AR-SAGAN effectively controlled the quality of the generated recipes by adjusting the preset control parameters. Especially when the feature extractor was trained with regressor loss using the real recipes and quality data, the AR-SAGAN had a better performance in data augmentation and quality prediction.

Our future work will focus on two parts, given as follows: first, a solution for the regression for unbalanced distributed data by improving the proposed method; and second, an extension of AR-SAGAN application by combining multivariable process modeling with other areas.

## Figures and Tables

**Figure 1 micromachines-13-00847-f001:**
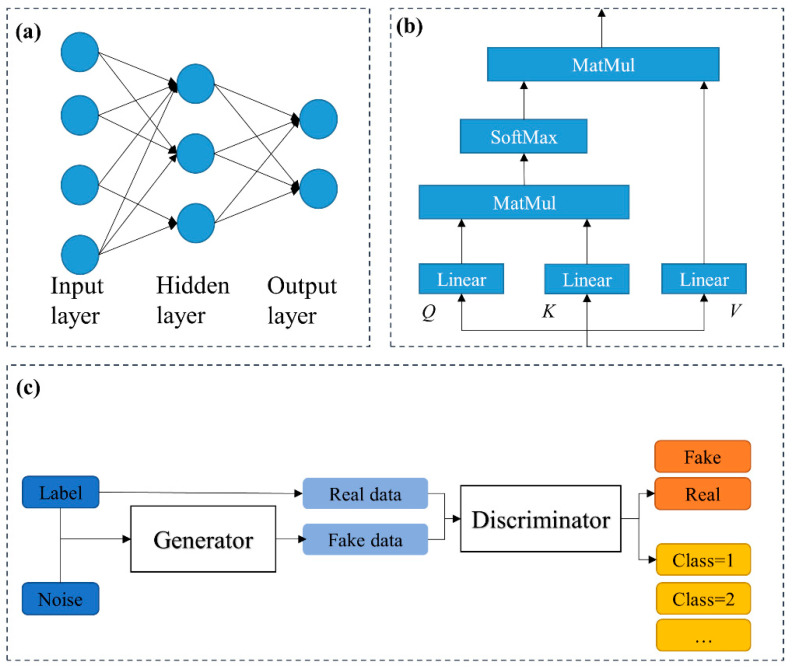
The schematic diagrams of (**a**) an ANN, (**b**) a self-attention mechanism, and (**c**) an auxiliary classifier GAN.

**Figure 2 micromachines-13-00847-f002:**
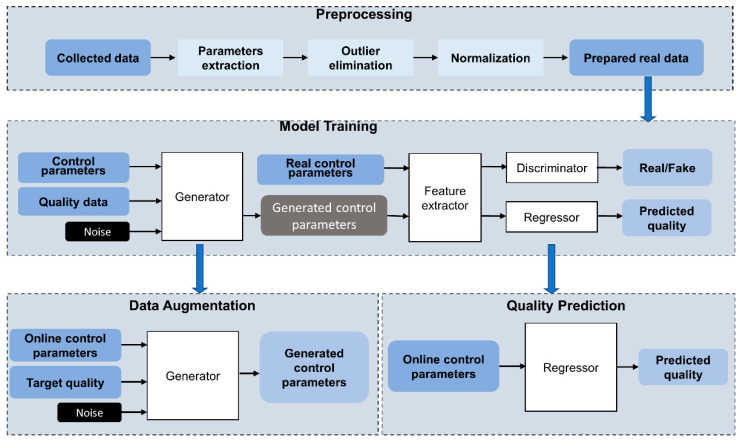
Overview of NCM process modeling using AR-SAGAN.

**Figure 3 micromachines-13-00847-f003:**
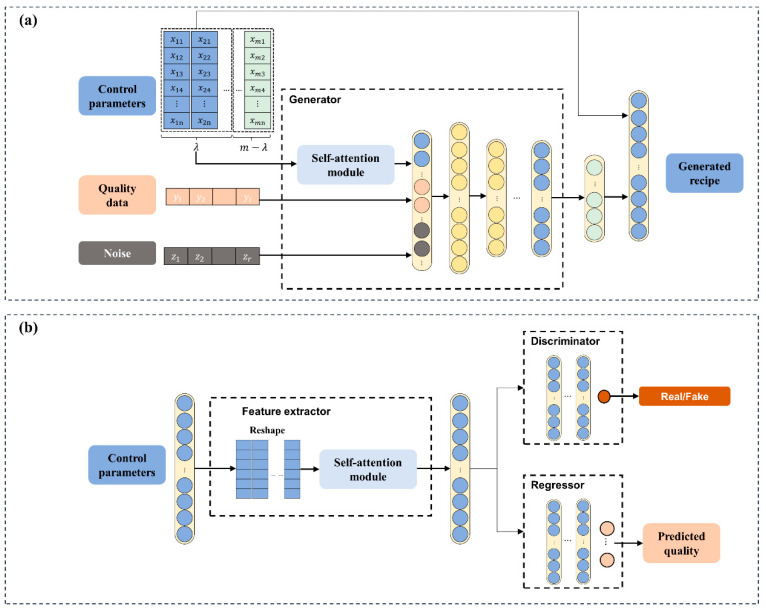
Specific architecture of AR-SAGAN includes (**a**) Generator, (**b**) Feature extractor, Discriminator and regressor.

**Figure 4 micromachines-13-00847-f004:**
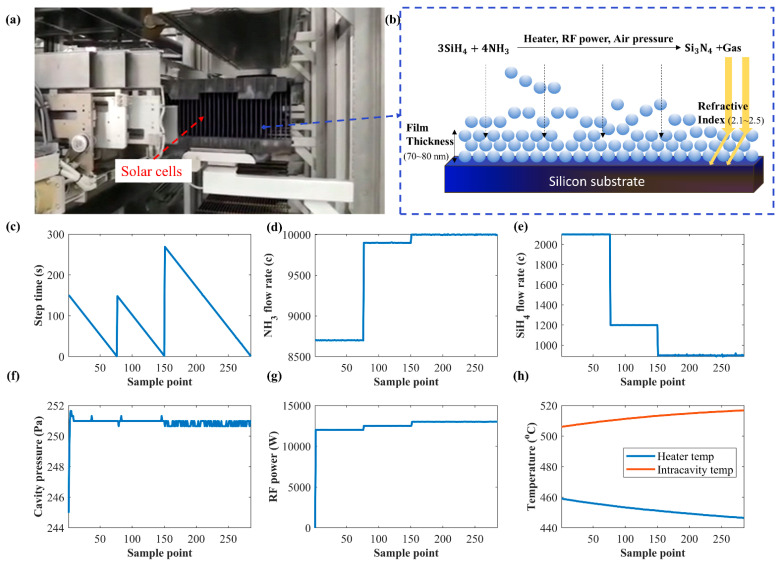
PECVD-based SiNx thin-film deposition process. (**a**) The real production process. (**b**) The schematic diagram of PECVD reaction process. (**c**–**h**) The data of control parameters collected from sensors.

**Figure 5 micromachines-13-00847-f005:**
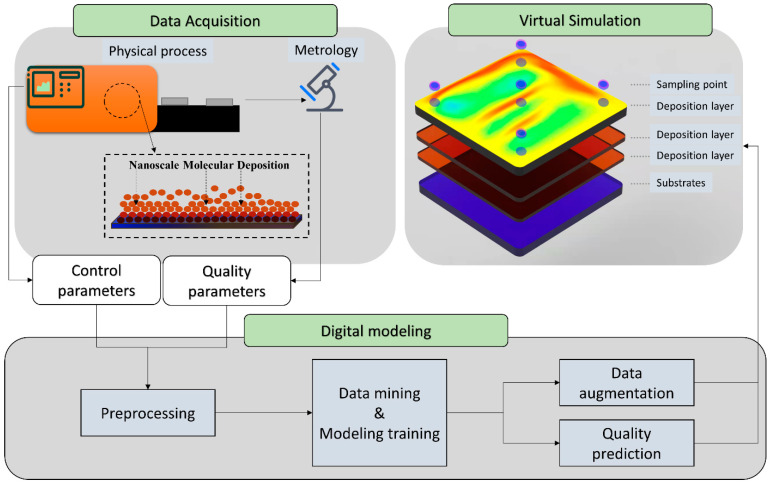
Schematic diagram of practical application in NCM process.

**Table 1 micromachines-13-00847-t001:** MAPE of generated control parameters under different training conditions.

Control Parameter	TC1	TC2	TC3
1	0.07353	0.05976	0.03904
2	0.03534	0.02876	0.03092
3	0.03125	0.02440	0.02318
4	0.01832	0.01118	0.00921
5	0.00370	0.00648	0.00096
6	0.00607	0.00614	0.00562
7	0.01228	0.01033	0.01106
8	0.01398	0.00966	0.00938
9	0.00921	0.00920	0.00976
10	0.01981	0.01504	0.01319
11	0.01209	0.01176	0.01114
12	0.01481	0.01546	0.01664
13	0.04192	0.05158	0.04206
14	0.01348	0.02008	0.02158
15	0.01931	0.01913	0.01945
16	0.03966	0.02920	0.03228
17	0.03939	0.03530	0.04110
18	0.03316	0.02360	0.02082
19	0.01759	0.02021	0.03367
20	0.00813	0.00608	0.00610
Mean ± Std.	0.0232 ± 0.0169	0.0207 ± 0.0146	0.0199 ± 0.0127

**Table 2 micromachines-13-00847-t002:** Predicted quality under different training conditions.

Quality Variable	Metrics	TC1		TC2		TC3	
		Train	Test	Train	Test	Train	Test
Thickness (nm)	MSE	2.0678	2.6034	1.7089	2.0579	1.6627	2.0111
	MAPE	0.0163	0.0185	0.0127	0.0148	0.0128	0.0149
Refractive index	MSE	6.588 × 10^−5^	6.232 × 10^−5^	6.775 × 10^−5^	6.186 × 10^−5^	7.072 × 10^−5^	6.194 × 10^−5^
	MAPE	0.0030	0.0028	0.0031	0.0029	0.0031	0.0029

**Table 3 micromachines-13-00847-t003:** Predicted quality under different methods.

Method	SVM		CGAN		AR-SAGAN	
	TN (nm)	RI	TN (nm)	RI	TN (nm)	RI
Train MSE	3.7215	0.0068	3.4107	8.5 × 10^−5^	1.6627	7.1 × 10^−5^
Test MSE	4.1665	0.0065	2.8082	8.6 × 10^−5^	2.0111	6.2 × 10^−5^

## Data Availability

Not applicable.

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
