# Peer review of "Self-Attention-Augmented Generative Adversarial Networks for Data-Driven Modeling of Nanoscale Coating Manufacturing"

_micromachines, 2022, doi:10.3390/mi13060847_

Round 1

Reviewer 1 Report

Nanoscale coating manufacturing (NCM) process modeling is an important way for monitoring and modulating coating quality. A novel auxiliary regression using self-attention augmented generative adversarial network (AR-SAGAN) is proposed in this paper. It is a meaningful work. Here are the suggestions before it could be accepted.

  1. The literature should be updated, more literature should be in recent three years.
  2. In the introduction, the disadvantages of the references should be summarized clearly to emphasize the importance of this work.
  3. The boundary conditions of the model should be given in details.
  4. There are too many formulas and symbols, so a nomenclature table is needed.
  5. In fig.2(b). I suggest that the thickness of film and matrix can be added. In Fig.2(c-h), the unit of ordinates should be added.
  6. The compared regression results of the AR-SAGAN and other methods are demonstrated in Table 3. But it is not clearly, how about drawing a figure to compare the results.
  7. The resolution of fig.1 should be improved.

Author Response

Thanks for your comments.

Reviewer 2 Report

In the proposed manuscript, entitled "Self-Attention Augmented Generative Adversarial Networks

for Data-driven Modeling of Nanoscale Coating Manufacturing" describes the results of a study devoted to the search for a model that improves the production processes of nanoscale coatings. The article is written in a professional language that is understandable to specialists. The logic of the narrative and the structuring of the text is present. There are several places that require minor correction.

1) Figure 1 is separated from the signature.

2) Line 125. The title is on a different page, relative to the text.

As for the content of the article, the proposed model gives the impression of a theoretical one, the mechanism of its application is not fully understood. The formulas given in most cases have a general appearance, devoid of specifics. The dimension of the quantities is not clear. Figure 1 and Figure 2 are general diagrams that serve for general reference, but do not illustrate the results of the study. Table 1 shows the results in a general way, without explanation. It is not clear how it is possible to predict the quality and generate the parameters mentioned in lines 92-94.

The presented manuscript contains a description of the approach to the organization of the study rather than the study itself. Perhaps it would be worth expanding the article with more information concerning the actual production of nanoscale coatings. The authors have done a lot of work, some of the results of which are not included in this article. I propose to write a detailed and understandable description of one individual example of the practical application of the proposed model, or add another chapter explaining the results, at the discretion of the authors.

Reviewer 3 Report

Ji et al. have dealt with the two existing problems in artificial intelligent model such as multivariable prediction of coated film and data augmentation of nanoscale coating manufacturing by proposing a novel auxiliary regression with self-attention augmented generative adversarial network. With the proposed model, the authors have resolved the above cited problems by three steps and verified the method by solar-cell antireflection coating experimental dataset. This work is an important attempt to solve the key problem in Al model towards effectively controlling the quality of generated recipes by adjusting the preset control parameters. Moreover the authors have envisaged on future works warranting solution of regression for unbalanced distributed data by improving the reported model and extending the proposed model's application to other areas of multivariable process modeling. Based on their well-conceived and appropriately executed work with lucid presentation, I recommend publication of this article after undergoing some minor corrections as shown below:

1. Some optimized parameters or quantitation data obtained by the proposed model to overcome the cited issues in Al model should be provided in the abstract.

2. All the abbreviations and symbols used in both tables and figures should be explained in footnotes and captions respectively.

3. At the end of the discussion, some statement clearly indicating how technical parameters obtained using the proposed model in this study was demonstrated to overcome the limitation of issues of with the existing traditional/conventional models/methods.
